# Preparation of Linear Actuators Based on Polyvinyl Alcohol Hydrogels Activated by AC Voltage

**DOI:** 10.3390/polym15122739

**Published:** 2023-06-19

**Authors:** Tarek Dayyoub, Aleksey Maksimkin, Dmitry I. Larionov, Olga V. Filippova, Dmitry V. Telyshev, Alexander Yu. Gerasimenko

**Affiliations:** 1Institute for Bionic Technologies and Engineering, I.M. Sechenov First Moscow State Medical University, Bolshaya Pirogovskaya Street 2-4, 119991 Moscow, Russia; aleksey_maksimkin@mail.ru (A.M.); dmitry.larionov0625@gmail.com (D.I.L.); borisovaolya@yandex.ru (O.V.F.); telyshev@bms.zone (D.V.T.); gerasimenko@bms.zone (A.Y.G.); 2Department of Physical Chemistry, National University of Science and Technology “MISIS”, 119049 Moscow, Russia; 3Institute of Biomedical Systems, National Research University of Electronic Technology, Zelenograd, 124498 Moscow, Russia

**Keywords:** polyvinyl alcohol, hydrogel, actuator, AC voltage, deformation, activation time

## Abstract

Currently, the preparation of actuators based on ionic electroactive polymers with a fast response is considered an urgent topic. In this article, a new approach to activate polyvinyl alcohol (PVA) hydrogels by applying an AC voltage is proposed. The suggested approach involves an activation mechanism in which the PVA hydrogel-based actuators undergo extension/contraction (swelling/shrinking) cycles due to the local vibration of the ions. The vibration does not cause movement towards the electrodes but results in hydrogel heating, transforming the water molecules into a gaseous state and causing the actuator to swell. Two types of linear actuators based on PVA hydrogels were prepared, using two types of reinforcement for the elastomeric shell (spiral weave and fabric woven braided mesh). The extension/contraction of the actuators, activation time, and efficiency were studied, considering the PVA content, applied voltage, frequency, and load. It was found that the overall extension of the spiral weave-reinforced actuators under a load of ~20 kPa can reach more than 60%, with an activation time of ~3 s by applying an AC voltage of 200 V and a frequency of 500 Hz. Conversely, the overall contraction of the actuators reinforced by fabric woven braided mesh under the same conditions can reach more than 20%, with an activation time of ~3 s. Moreover, the activation force (swelling load) of the PVA hydrogels can reach up to 297 kPa. The developed actuators have broad applications in medicine, soft robotics, the aerospace industry, and artificial muscles.

## 1. Introduction

Generally, actuators are devices that convert applied energy into motion. Unlike traditional actuators such as piezoelectric ceramic actuators, there is currently increased interest in polymeric actuators due to their unique properties such as a lightweight design, simplicity, noiselessness, biodegradability, a fast response time, and good mechanical properties [1,2,3,4]. When polymeric actuators are exposed to external stimuli such as chemical reactions (changes in pH), electric current, changes in humidity, light, changes in temperature, magnetic fields, etc., they can respond by changing their size or shape. These materials can return to their original shape once the external influence is removed [5,6]. One of the most important and interesting types of actuators are electroactive polymers (EAP), which can be deformed by applying an external electric field. Their characteristics, such as a lightweight design, quietness, biodegradability, biocompatibility, and good mechanical properties, especially under high loads, make them attractive for applications such as sensors, soft robotics, and artificial muscles [3,4,5,6,7,8]. EAPs are polymers that change their mechanical or optical characteristics when they are subjected to an electrical current. They do so with precise control of the conversion of electrical to mechanical energy and vice versa, making them a preferred choice for electromechanical actuator applications [9]. Depending on the mechanism of conduction, electroactive polymers can be classified into materials with electrical conductivity, resulting from the mobility of electrons or conductive particles, and materials with ionic conductivity, such as hydrogels [10]. The basic principle of operation for ionic EAPs is the diffusion of ions, and the change in the size and shape of the actuator depends on the mobility and diffusion of ions in the ionic liquids of the electrolyte. When an electric field is applied, anions and cations begin to diffuse towards the anode and cathode, respectively, which leads to swelling, compression, and bending of the actuator [3,4,10]. However, ionic EAPs are activated using DC voltage and require a low voltage for activation. They also exhibit bistability (they have two stable equilibrium states). The disadvantages of ionic EAPs include the need to maintain constant humidity, the potential for electrolysis above a certain voltage, which can lead to irreversible damage to the material, slow activation, low strength, and difficulties in maintaining constant deformation when a constant voltage is applied [11]. 

Polyvinyl alcohol and poly(2-acrylamide-2-methyl-1-propanesulfonic acid) can form gels and are used as ionic EAPs. Depending on the solvents used, gels can be classified as aqueous (hydrogels) or non-aqueous ionic gels, which can be prepared by swelling polymer networks using organic electrolyte solutions that exhibit physical interactions within the polymer networks [12,13,14]. Polyvinyl alcohol (PVA) is a well-known polymer that forms gels and is used as an ionic EAP. PVA is a biodegradable, biocompatible, and hydrophilic synthetic polymer with excellent adhesive properties [15,16,17,18]. PVA has been widely applied in many fields such as food packaging, paper, textiles, wastewater treatment, and biomedical applications [19,20]. PVA, being a semicrystalline polymer, exhibits the property of the shape memory effect [21]. PVA, similar to other shape memory polymers, consists of two parts: switching segments (soft segments) and net points (hard segments). The crystalline domains serve as connection points, whereas the amorphous domains act as switching segments. The shape memory effect of PVA can vary depending on the degree of crosslinking [22]. In the reference [23], a water-induced shape memory polymer based on polyvinyl alcohol (PVA) was prepared by introducing graphene oxide. The authors demonstrated that the strong hydrogen bonding interaction between PVA and graphene oxide allowed for the formation of additional physically cross-linked points in PVA, significantly improving its shape memory properties. Moreover, the authors explained that the water-induced shape recovery was due to the decrease in glass transition temperature and storage modulus, which can be explained by the swelling plasticizing effect of water on PVA, as evidenced by the obvious increase in volume. In reference [24], polyvinyl alcohol (PVA) films containing a few layers of graphene (FLG) were prepared by co-mixing aqueous colloids and casting. The authors revealed that the addition of graphene to PVA resulted in enhanced mechanical properties and electrical conductivity. They reported that the tensile modulus could be increased by up to 114%, and the tensile strength by up to 60%. The highest electrical conductivity achieved was 10–3 S/cm with a 3% FLG content.

In reference [25], the authors prepared a bilayered hydrogel actuator by coating a borax cross-linked N1,N1-diethylethane-1,2-diamine (DEEDA)-modified polyvinyl alcohol (PVA) microgel (PVA–DEEDA–borax) onto a polyacrylic acid-coated biaxially oriented polypropylene (BOPP) substrate. This type of actuator can be activated by heat, moisture, or IR light stimuli. The authors demonstrated that their actuators exhibited rapid, reversible, and sustainable bidirectional self-rolling deformation, achieved through a swelling/deswelling process, enabling the formation of a 2D/3D tube, wave, and cross shapes through deliberate assembly of the actuators. In reference [26], the authors prepared a hydrogel actuator for electromechanical actuator applications based on oxidized multiwalled carbon nanotubes (oxidized-MWNT)/polyvinyl alcohol (PVA). They demonstrated that their prepared actuator can be activated by applying different DC voltages from 2 to 10 V in a liquid electrolyte (tetrabutylammonium/tetrafluoroborate (TBA/TFB) in acetonitrile). The authors showed that the stress generated by the actuator increases with increasing the amplitude of the applied voltage and can reach 1.8 MPa. In the reference [27], a hydrogel actuator based on multiwalled carbon nanotube (MWNT) and polyvinyl alcohol was prepared. The authors presented that when applying a DC electric field, their actuator bends due to the change in osmotic pressure under the DC electric field. The authors illustrated that with the reversal of the polarity of the applied DC electric field, the mobile ions both outside and inside the hydrogel actuator move to the opposite direction, resulting in the reverse bending of the actuator. They showed that the activation time for the actuator bending was approximately 80 s. In reference [28], the authors prepared an electroactive hydrogel based on polyvinyl alcohol (PVA) and cellulose nanocrystal (CNC) using a freeze–thaw technique. They demonstrated that by applying a high DC voltage in the range of 0.6–1.6 kV at a frequency of 0.1 Hz to the prepared hydrogels in an aqueous swollen state in DI water, the displacement of the activated actuator increased with increasing applied actuation voltage. The authors explained that their findings are associated with the interfacial polarization between CNCs and the PVA polymer matrix, which leads to improved electroactive behavior of the material due to the increased dielectric properties of the hydrogel. In the reference [29], the authors prepared PVA/tendon hydrogels as anti-freezing gels with good mechanical properties even at low temperatures for use as hydraulic actuators and ionic conductors. The authors reported that their actuators were activated using air flow or a dimethyl sulfoxide (DMSO)/H_2_O mixture, and these actuators exhibited a fast response rate and high actuation force.

Generally, the ability of polymeric hydrogels to absorb water arises from hydrophilic functional groups, which are connected to the polymeric backbone, whereas their dissolution resistance is related to the cross-links between polymeric chains and the absence of capillary channels [30]. In the reference [31], the authors prepared fast water transport carbon nanotube (CNT)/poly vinyl alcohol (PVA) hydrogels. They demonstrated that the prepared double-network structure of the PVA/CNT hydrogel provides hierarchical micro/nanochannels and strong capillary forces, resulting in water transport throughout the hydrogel at a rate of up to ~15.8 g. g^−1^s^−1^. Moreover, the authors illustrated that the prepared hybrid hydrogels show a reversible absorbing/shrinking behavior, excellent water absorption capacity (216 g. g^−1^), and high adaptability to boiling water.

In contrast to the circular motion of a traditional electric motor, a linear actuator generates motion in a straight line. Linear actuators are used in machine tools, industrial machinery, computer peripherals such as disk drives and printers, valves, dampers, artificial muscles, and numerous other applications requiring linear motion. For example, cylinders, whether hydraulic or pneumatic, naturally provide linear motion. In reference [32], the authors created a system that combines the benefits of rigid and soft robotics by creating an affordable, dependable, muscle-like linear soft actuator used antagonistically to operate a stiff 1-DoF joint. In reference [33], the authors prepared reverse pneumatic artificial muscle (rPAM) using silicone rubber that is radially constrained by symmetrical double-helix threading. Moreover, spiral artificial muscles based on polyethylene and nylon fibers are also considered a type of linear actuator capable of reversible contraction/extension deformation under heating/cooling cycles [34,35].

As mentioned in the previous literature review, the activation of actuators based on electroactive PVA hydrogels occurs using DC voltage, and these actuators have a slow activation response. In this article, a new approach for activating PVA-based actuators using AC voltage is proposed. The suggested actuators are activated through a swelling/shrinking mechanism caused by the local vibration of ions, without any movement towards the electrodes. This is in contrast to ionic hydrogels that are activated by DC voltage, which leads to material heating and the transformation of water molecules within the hydrogel into a gaseous state, leading to actuator swelling. To achieve linear movement for the proposed actuators, two types of external reinforcement materials (spiral weave and fabric woven braided mesh) were used. The study investigated actuator deformation, activation time, and efficiency, considering factors such as PVA concentration, applied voltage, frequency, and load.

## 2. Materials and Methods

### 2.1. Materials

Polyvinyl alcohol (PVA) with an average molecular weight of 105,000 g/mol and a hydrolysis degree of 99% was purchased from Ruskhim Ltd. (Moscow, Russia). Sodium tetraborate (borax) was used as a crosslinking agent for the PVA hydrogels, and distilled water was used as the solvent for the preparation. Latex balloons were used as an elastomeric shell for the preparation of the actuators based on the PVA hydrogels. A nylon 6 fiber with a diameter of 0.5 mm was used to prepare a spiral weave as an external reinforcement material for the first type of linear actuators. Fabric woven braided mesh made of polyethylene terephthalate with an internal diameter of 6 mm and the ability to stretch twice was used as an external reinforcement material for the second type of linear actuators. Copper electrical wires were used as electrodes.

### 2.2. Preparation of PVA Hydrogels

Using a magnetic stirrer, a PVA solution in distilled water was prepared by heating the solution to a temperature of 140 °C for 1 h. Then, a borax solution in distilled water was added to the PVA solution and stirred at the same temperature for an additional 1 h. Subsequently, the PVA/borax solution was subjected to vacuuming to remove air bubbles from the prepared hydrogels. The resulting solutions were stored in a refrigerator at 4 °C for further use.

In order to study the influence of PVA concentration on actuator deformation and activation time, hydrogels with varying PVA content were prepared. A total of 100 mL of distilled water was used for the preparation of all PVA/borax solutions. A borax content of 2 wt.% of the PVA mass content was added to all hydrogel composites. Table 1 shows the PVA and borax contents in the prepared hydrogels. Figure 1 shows the chemical reaction between PVA and borax.

### 2.3. Preparation of Hydrogel Actuators

A stretchable elastomeric shell with a length of 50 ± 5 mm and an inner diameter of 6 ± 0.5 mm was filled with 2 mL of PVA hydrogel. Two copper wires were used as electrodes. The length of each electrode inside the shell was 25 ± 1 mm. Figure 2a shows a schematic design of the PVA hydrogel-based actuators. The stretchable elastomeric shell was reinforced by two types of external reinforcement, as shown in Figure 2b,c. The internal diameter of both external reinforcements was 6 ± 0.5 mm.

### 2.4. Testing Procedures

#### 2.4.1. Measurements of Electrical Resistance Using the Four-Probe Method

A conductivity test of the PVA hydrogels was performed on a 4-point probe resistivity measurement system (RLC meter AKIP-6112/2) with an electrical capacitance range of 0.01 nF to 10 F. A total of 3 measurements for each hydrogel composite were carried out for frequency values of 50 and 500 Hz under a voltage of 2 V.

#### 2.4.2. Actuation Tests

For the investigation of the actuation deformation, response time, and displacement values, different AC voltage values ranging from 90–200 V, two different frequency values (50 and 500 Hz), and two loads of 50 and 100 g were applied to all types of actuators based on PVA hydrogels (see Appendix A). An AC power source (MATRIX APS-7100 AC Power Source) with an operating voltage range of 0–310 V and a frequency range of 45–500 Hz was used. Depending on the inner diameter of the stretchable elastomeric shell (actuator area), the applied load of 50 g corresponds to approximately 20 kPa, and the applied load of 100 g corresponds to approximately 40 kPa.

#### 2.4.3. Measurement of Actuator Deformation

To investigate the contraction/extension values of the prepared actuators, the high-performance laser distance sensor Wenglor YP11MGVL80 (Wenglor sensoric GmbH, Tettnang, Germany) was used. This sensor has the following characteristics: linearity—0.5%, measuring range—50 mm, resolution—20 um, shown in Figure 3.

To study the influence of applied load on the deformation and response time of the prepared actuators, two loads of 50 g (~20 kPa) and 100 g (~40 kPa) were applied to the actuators. To collect and process the obtained data, a combination of a data acquisition board with a USB interface, LA-2USB-14 (LLC Rudnev-Shilyaev, Moscow, Russia), and PowerGraph software 3.3.11 (DISoft, Zarechny, Russia) were used. The ADC included in LA2-USB-14 has the following specifications: 14-bit resolution, and a maximum discrete frequency of 400 kHz (in this work, the sampling rate was 1 kHz). The activation time of the actuators was calculated as the duration between two moments: the first moment being the application of voltage and the second being the peak deformation.

#### 2.4.4. Measurement of Activation Force of PVA Hydrogels

A Single Point Parallel Beam Loadcell (SUP 1) (ShengWanJiang Transducer Technology (Shenzhen) Co., Ltd., Shenzhen, China) was used to measure the activation stress of the PVA hydrogels. The loadcell had the following specifications: largest measurement limit—5 kg, zero balance ± 5%, non-linearity—0.05%, complex error—0.05%.

A total of 2 mL of PVA hydrogel was placed in a syringe with a volume of 5 mm. Figure 4 shows a schematic design of the setup used for the measurement of hydrogel efficiency. 

To calculate the efficiency of PVA hydrogels, the design presented in Figure 4a was employed. A load of 50 g was applied to the top of the syringe. The actuation deformation and activation time were calculated using a laser distance sensor as described in Section 2.4.3. The AC voltage was set to 110 V, frequency to 500 Hz, and the load was 50 g (~12 kPa). A thermocouple (TC-K-TYPE-1M (−50–+204 °C) (TP-01)) was used to measure the temperature of the PVA hydrogel.

The efficiency of both actuators and PVA hydrogels was calculated based on the following equations:ƞ = P_2_/P_1_ × 100, % (1)
where ƞ represents efficiency, P_1_ represents the supplied electrical power, and P_2_ represents the useful mechanical power generated by the actuator.

P_1_ was calculated using the equation:P_1_ = V. I, watt (2)
where V represents the voltage (volts) and I represents the current (Amps).

P_2_ was calculated using the equation:P_2_ = m. a. L/t, watt (3)
where m represents the applied load (kg), a represents acceleration (9.81 m/s^2^), L represents displacement (m), and t represents activation time (s).

It should be noted that the useful mechanical power generated by the actuator was calculated using the determined values of contraction/extension deformation presented in Appendix A for 1 m of actuator length. 

## 3. Results

Since alternating current (AC) is a cyclic voltage that switches periodically between positive and negative values, causing the direction of electricity flow to change accordingly, the ions in hydrogels do not move towards the electrodes but will be subjected to local vibrations, resulting in the heating of the hydrogel.

When an AC voltage is applied, the hydrogel undergoes heating, leading to the swelling of the samples. This phenomenon can be explained by the interaction of molecular bonds with the network structure, as depicted in Figure 5 (see Appendix A). Under normal room conditions (AC voltage is off and temperature is 25 °C), the hydrogel has a minor swelling ratio, which is related to the compressed network structure formed by hydrogen bonds. This is referred to as the “shrinking and bonding status”. By applying the AC voltage, the hydrogel heats up, leading to a weakness in the hydrogen bonds, which in turn leads to a stretching in the twisted polymer molecular chain. This will lead to a volumetric contraction of the PVA hydrogel with a decrease in its viscosity. When the temperature is higher than 95 °C, the hydrogen bonds between the water molecules and PVA in the hydrogel network will break, and the water molecules will be transformed into a gaseous state, causing an increase in the volume of the elastomeric shell that covers the hydrogel. When the voltage is turned off, the evaporated water will be transformed into a liquid state and reabsorbed in the hydrogel, which leads to the restoration of the original shape of the elastomeric shell [36,37]. In reference [38], polyvinyl alcohol gel was investigated as a temperature-sensitive polymeric gel. They demonstrated that PVA hydrogels have a critical transition temperature (with H_2_O) of 37 °C. They illustrated that when the hydrogel temperature is higher than 37 °C, the hydrogel can easily discharge the water molecules, whereas at a lower temperature, the hydrogel can strongly adsorb the water molecules.

Based on the state diagram for water in the gas–liquid phase [39], the latent heat of vaporization (L_v_) is equivalent to 2501 J/g, the specific heat capacity for the liquid (C_p_^liquid^—the energy required to increase the temperature by one degree Celsius until it reaches 100 °C) is 4.218 J/g.°C, and the specific heat capacity for the gas (C_p_^gas^) is 1.901 J/g.°C. This means that each gram of water requires 2501 J/s (watt) to convert it from a liquid to a gas. Considering that the average activation time of the actuator is 3 s, the required power for vaporization is approximately 7.5 watts for each gram of water.

In addition, the forward osmosis process plays an important role in the swelling/contraction processes of hydrogels. It involves the transport of water across a semipermeable membrane (hydrogel) from the side with a higher water chemical potential to the other side with a lower water chemical potential [40]. Thermodynamically, the polymer–hydrogel network structure serves multiple functions as a solute, osmotic membrane, and pressure-generating device. In the case of ionic hydrogel networks, the swelling forces are greatly increased due to the localization of charges on the polymer chains, similar to a polymer solution where ionic groups are dissolved. At the equilibrium of swelling, the total change in free energy is minimized, resembling a system in which the chemical potential of each mobile species becomes equal in coexisting phases. Therefore, the expansion and contraction behavior of hydrogels in response to temperature changes is also related to the forward osmosis hydration/dehydration processes [39,41,42].

Figure 6, Figure 7 and Figure 8, Appendix A, and Appendix A show the results of the overall extension, contraction, and activation time of PVA hydrogels under varying AC voltage, frequencies, and loads.

As can be seen in Figure 6, Figure 7 and Figure 8 and Appendix A, increasing the PVA content, voltage, and frequency improved the deformation of the actuators and reduced the activation time. It should be noted that since the increase in the frequency values led to an increase in the deformation values, the increase in the frequency values had a significant effect on the activation time of the actuators, which led to a decrease in the activation time by increasing the frequency tenfold. Moreover, as can be seen in Appendix A, the performance of the actuators was improved by increasing the PVA content. This can also be related to the improvement in the electrical conductivity of the PVA hydrogels (Table 2). In addition, it should be noted that for the P7B2 and P10B2 hydrogels, there was not a significant difference in the deformation values, whereas the P7B2 hydrogels had better activation times in comparison with the P10B2 hydrogels. This can be related to the number of hydrogen bonds. In other words, since the PVA content is higher in the P10B2 hydrogels than in the P7B2 hydrogels, the number of hydrogen bonds will be higher, which in turn means that the required time for heating of the hydrogel and breaking of the hydrogen bonds will be higher for the P10B2 hydrogels. From a technical perspective, it should be noted that PVA hydrogels with a PVA content of less than 5% are more fluid, whereas those with a PVA content of more than 7% are too viscous. Therefore, PVA hydrogels with a PVA content of 7% (P7B2 hydrogels) are considered the best for the preparation of actuators. As can be seen in Appendix A and Figure 7 and Figure 8, when applying an AC voltage of 200 V, a frequency of 500 Hz, and a load of 20 kPa to the P7B2 hydrogels, the stretching/shrinking deformation was ~60% and the activation time was ~3 s for the external reinforcement of spiral weave. The stretching/shrinking deformation was ~22% and activation time was ~3 s for the external reinforcement of fabric woven braided mesh (see Appendix A). The limited deformation of the actuator based on PVA hydrogel reinforced by fabric woven braided mesh can be attributed to the fact that the design of this type of reinforcement can only stretch up to twice its original length (2:1).

Table 3, Table 4, Table 5 and Table 6 show the electrical efficiency of the actuators based on PVA hydrogel. The electrical efficiency of the actuators based on PVA hydrogel was calculated using equations (1–3). As can be seen, the electrical efficiency of the actuators was generally low. The actuator based on PVA hydrogel reinforced by spiral weave had a higher electrical efficiency (up to 1.58%) than the actuator based on PVA hydrogel reinforced by fabric woven braided mesh (up to 0.86%). This difference is related to the lower deformation capability of the actuator based on PVA hydrogel reinforced by fabric woven braided mesh.

Table 7 and Figure 9 show the obtained force of the PVA hydrogel (P7B2) depending on the measurement procedure, which is described in Section 2.4.4. As can be seen, by increasing the values of AC voltage and frequency, the activation force was increased by up to 2.375 kg (297 kPa). Additionally, the required time for obtaining the maximum activation force was decreased from 4.18 s to 1.5 s under an AC voltage of 200 V and a frequency of 500 Hz. The temperature changes during the activation process for the PVA hydrogel were found to be within the range of 90–110 °C (see Appendix A). Moreover, by increasing the values of AC voltage and frequency, the amount of water vapor formed was increased, while the formation time was decreased.

Table 8 shows the activation displacement, time, and electrical efficiency of PVA hydrogel P7B2 under an AC-voltage of 110 V and a frequency of 500 Hz. As can be seen, and in comparison with the data in Table 3 and Table 4, the PVA hydrogel exhibited a higher electrical efficiency in comparison with the prepared actuators under the same conditions. However, it should be noted that the electrical efficiency of the PVA hydrogel can be increased by increasing the AC voltage. Under the conditions of 110 V, 500 Hz, and a load of 50 g, the PVA hydrogel can reach an efficiency of 2.44%.

## 4. Conclusions and Remarks

In general, the activation method for actuators based on ionic electroactive hydrogels involves the diffusion of ions through the application of DC voltage. This results in actuator swelling as anions and cations move towards the anode and cathode, respectively. However, these DC-voltage-activated actuators have a slow response, low electromechanical coupling efficiency, and a low actuation force. In this article, a new activation mechanism for actuators based on polyvinyl alcohol (PVA) hydrogels using AC voltage was proposed. This activation mechanism depends on the heating of the hydrogel under AC voltage, causing the water molecules within the hydrogel to transform into a gaseous state, leading to swelling of the actuator. Two types of linear actuators were prepared using two types of reinforcement (spiral weave and fabric woven braided mesh) that enable the conversion of swelling movement into linear movement. The deformation of the actuators and their activation time during extension/contraction cycles were investigated as a function of the AC voltage, frequency, and load. It was found that by applying an AC voltage of 200 V, a frequency of 500 Hz, and a load of ~20 kPa, the overall extension of the actuators reinforced by spiral weave could reach more than 60%, with an activation time of ~3 s. Similarly, the overall contraction of the actuators reinforced by fabric woven braided mesh could reach more than 20% with an activation time of ~3 s. It should be noted that in general, increasing the AC voltage, frequency, and polymer concentration led to increased overall extension/contraction and decreased activation time, as shown in Figure 10.

However, the electrical efficiency of the prepared actuators is considered low, measuring approximately 1.5% for the actuators reinforced by spiral weave and approximately 0.8% for the actuators reinforced by fabric woven braided mesh. It was found that the activation force (swelling load) of the PVA hydrogels could reach up to 297 kPa.

Due to the good contraction/extension values, fast response, and high activation displacement, the actuators studied in this article can be widely used in medicine, soft robotics, the aerospace industry, and as artificial muscles. However, it should be noted that these prepared actuators require some time (approximately 5–10 s) to be heated and activated at the beginning of the activation. They can hold the level of deformation and force for approximately 20 s before starting to relax. It also should be noted that the recovery time strongly depends on the applied load, the number of contraction/extension cycles, and the type of reinforcement. It was found that after many contraction/extension cycles, the recovery time was significantly increased by up to two times compared with the activation time. This could be attributed to the high heat capacity of water, which warrants further investigation to effectively manage the recovery time. Moreover, the high AC voltage requirement for the activation process requires further investigation in order to reduce the voltage requirement.

## Figures and Tables

**Figure 1 polymers-15-02739-f001:**
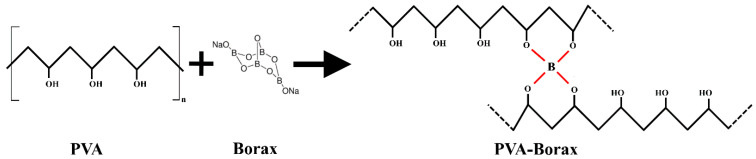
Chemical reaction of PVA and borax.

**Figure 2 polymers-15-02739-f002:**
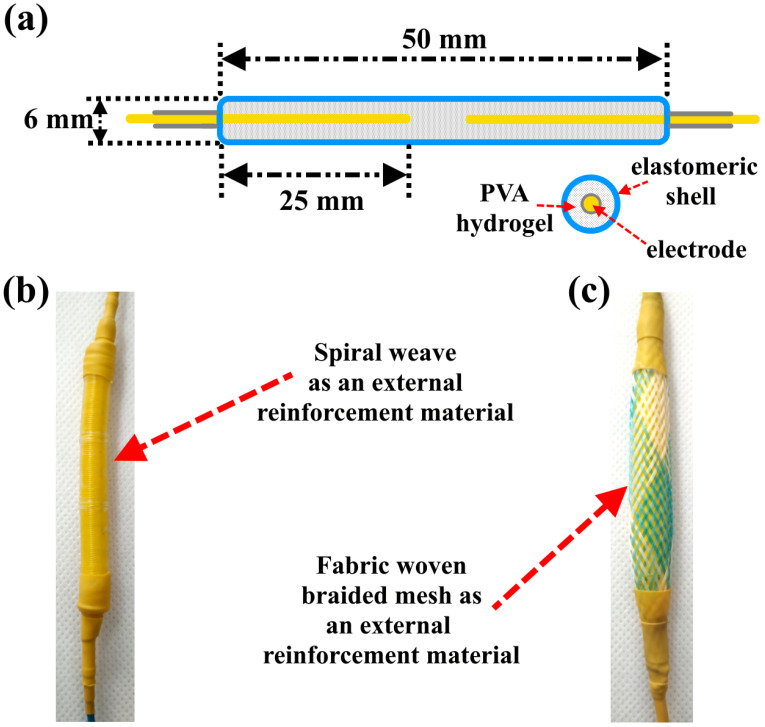
(**a**) Schematic design for an actuator based on PVA hydrogel, (**b**) actuator reinforced by a spiral weave, (**c**) actuator reinforced by a fabric woven braided mesh.

**Figure 3 polymers-15-02739-f003:**
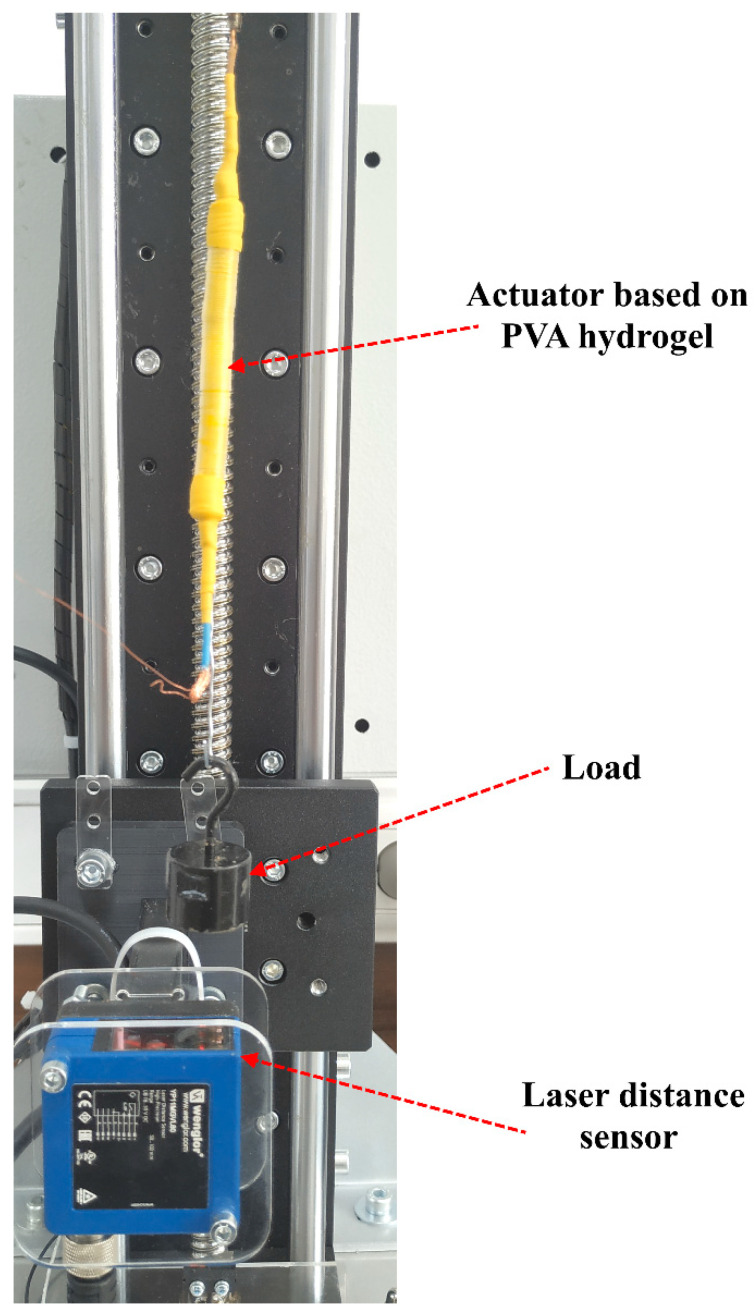
Measurement of actuator deformation using a laser detector.

**Figure 4 polymers-15-02739-f004:**
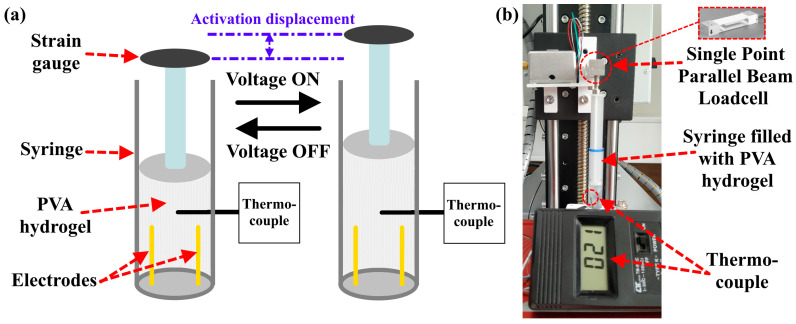
(**a**) Schematic design for the measurement of activation displacement of PVA hydrogel, (**b**) the used device.

**Figure 5 polymers-15-02739-f005:**
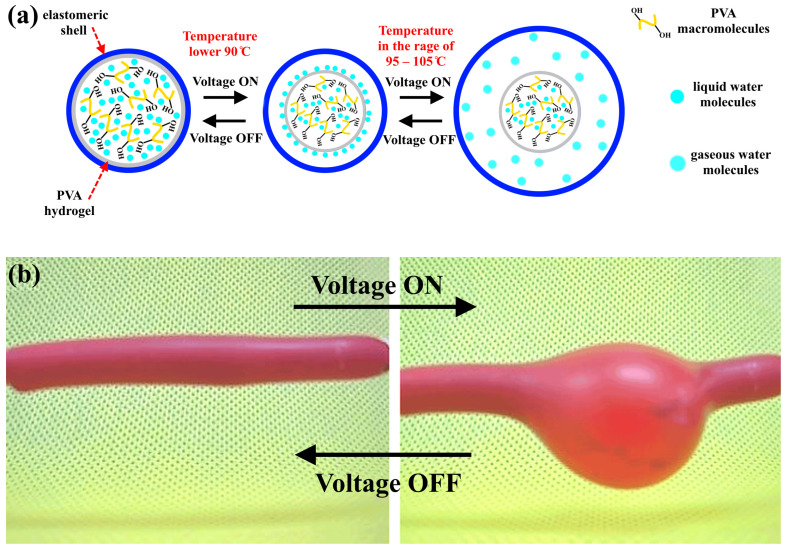
(**a**) Schematic diagram of the activation mechanism of PVA hydrogel by AC voltage. (**b**) Photo showing the swelling/shrinking mechanism of the actuator under AC voltage.

**Figure 6 polymers-15-02739-f006:**
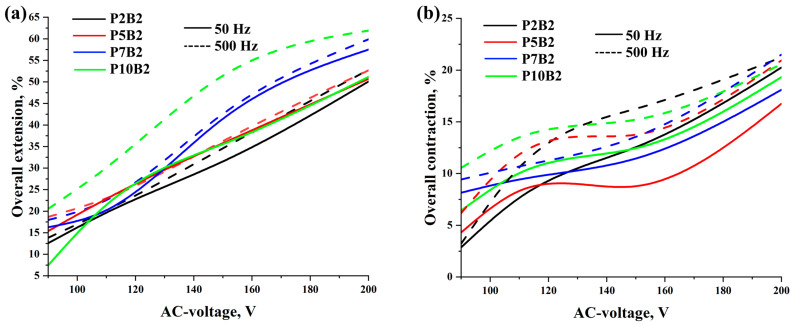
Overall extension/contraction of PVA hydrogels reinforced by (**a**) spiral weave and (**b**) fabric woven braided mesh under different values of AC voltage, frequency, and a load of 20 kPa. Standard deviations are presented in Appendix A.

**Figure 7 polymers-15-02739-f007:**
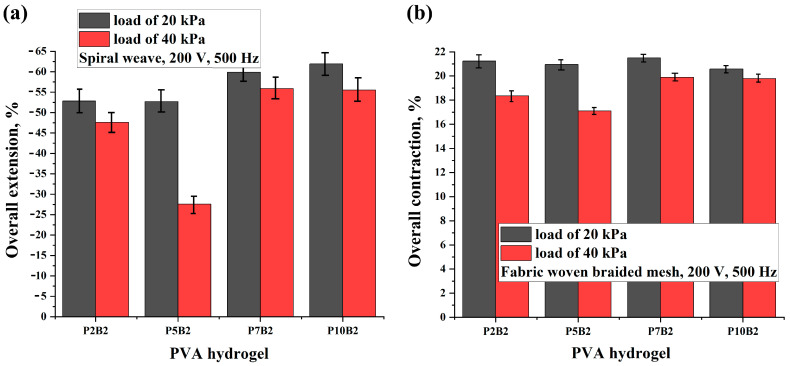
Overall extension/contraction of PVA hydrogels reinforced by (**a**) spiral weave and (**b**) fabric woven braided mesh under different load values, an AC voltage of 200 V, and a frequency of 500 Hz.

**Figure 8 polymers-15-02739-f008:**
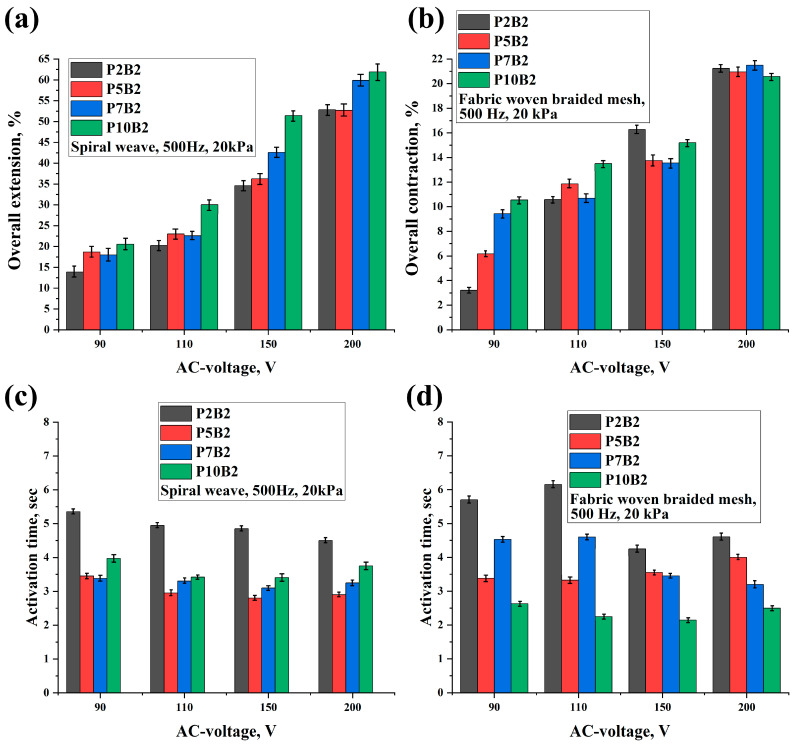
Overall extension (**a**) and contraction (**b**) of PVA hydrogels and their activation times (**c**,**d**) under a frequency of 500 Hz and load of 20 kPa at different values of AC voltage.

**Figure 9 polymers-15-02739-f009:**
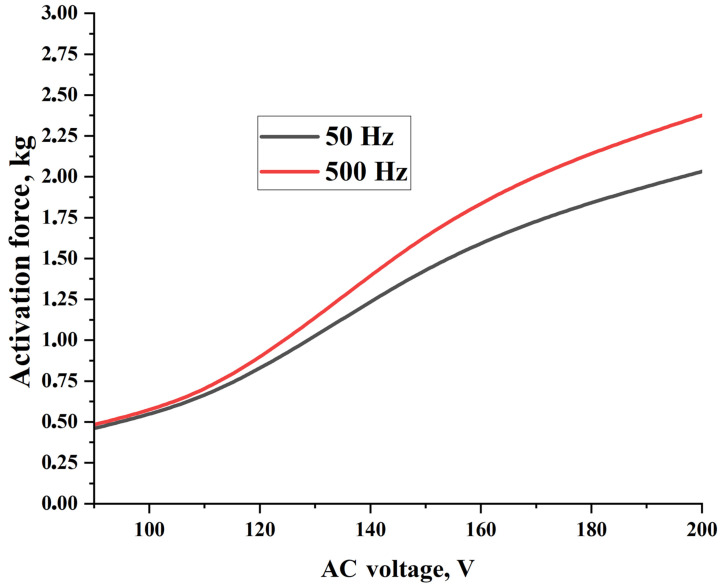
Achieved force by PVA hydrogel P7B2 under different values of AC voltage and frequency.

**Figure 10 polymers-15-02739-f010:**
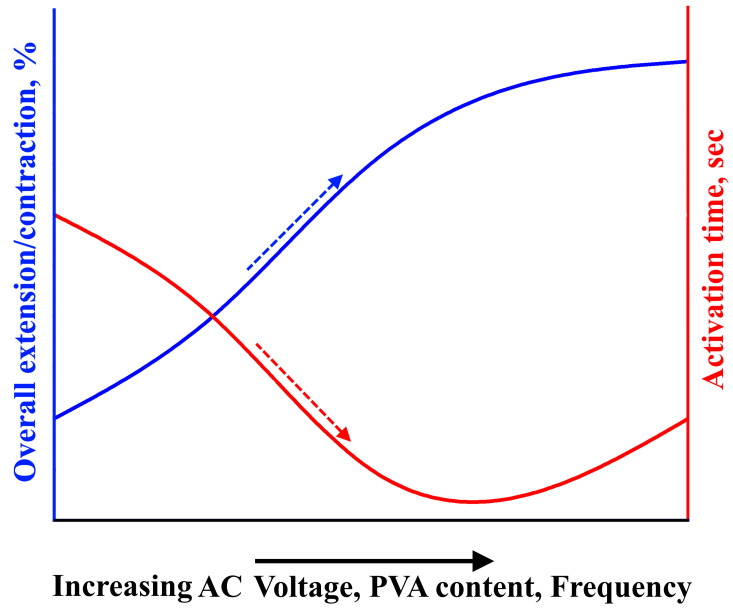
Effect of AC voltage, frequency, and PVA content on overall extension, contraction, and activation time of the actuators.

**Table 1 polymers-15-02739-t001:** PVA and borax contents in the prepared hydrogels.

Material Number	Material Code	PVA Concentration, wt. % in 100 mL of DI Water	PVA Concentration, mol/L	Borax Concentration, mol/L
1	P2B2	2	0.00019	0.00105
2	P5B2	5	0.00048	0.00262
3	P7B2	7	0.00067	0.00367
4	P10B2	10	0.00095	0.00525

**Table 2 polymers-15-02739-t002:** Electrical resistance of PVA hydrogels using the four-probe method.

Material	P2B2		P5B2		P7B2		P10B2	
Frequency, Hz	50	500	50	500	50	500	50	500
Electrical resistance, Ω	1851 ± 8	1625 ± 5	1641 ± 8	1347 ± 5	1352 ± 11	1102 ± 8	1171 ± 10	969 ± 2

**Table 3 polymers-15-02739-t003:** Electrical efficiency of the actuator based on PVA hydrogel under conditions of 110 V, 500 Hz, 20 kPa, with spiral weave reinforcement.

Material	P2B2	P5B2	P7B2	P10B2
Current, Amps	0.020 ± 0.02	0.022 ± 0.02	0.024 ± 0.03	0.031 ± 0.02
Power, Watt	2.20 ± 0.4	2.42 ± 0.7	2.64 ± 0.9	3.41 ± 0.8
Power of actuator, Watt	0.033	0.038	0.034	0.043
Efficiency, %	1.53	1.58	1.27	1.27

**Table 4 polymers-15-02739-t004:** Electrical efficiency of the actuator based on PVA hydrogel under conditions of 110 V, 500 Hz, 20 kPa, with fabric woven braided mesh reinforcement.

Material	P2B2	P5B2	P7B2	P10B2
Current, A	0.020 ± 0.02	0.022 ± 0.02	0.024 ± 0.03	0.031 ± 0.02
Power, Watt	2.20 ± 0.4	2.42 ± 0.7	2.64 ± 0.9	3.41 ± 0.8
Power of actuator, Watt	0.008	0.017	0.011	0.029
Efficiency, %	0.38	0.72	0.43	0.86

**Table 5 polymers-15-02739-t005:** Electrical efficiency of the actuator based on PVA hydrogel under conditions of 200 V, 500 Hz, 20 kPa, with spiral weave reinforcement.

Material	P2B2	P5B2	P7B2	P10B2
Current, A	0.024 ± 0.02	0.025 ± 0.02	0.031 ± 0.03	0.035 ± 0.02
Power, Watt	4.80 ± 0.3	5.10 ± 0.2	6.2 ± 0.1	7.1 ± 0.3
Power of actuator, Watt	0.058	0.067	0.090	0.081
Efficiency, %	1.21	1.31	1.45	1.14

**Table 6 polymers-15-02739-t006:** Electrical efficiency of the actuator based on PVA hydrogel under conditions of 200 V, 500 Hz, 20 kPa, with fabric woven braided mesh reinforcement.

Material	P2B2	P5B2	P7B2	P10B2
Current, A	0.024 ± 0.02	0.025 ± 0.02	0.031 ± 0.03	0.035 ± 0.02
Power, Watt	4.80 ± 0.3	5.10 ± 0.2	6.2 ± 0.1	7.1 ± 0.3
Power of actuator, Watt	0.040	0.041	0.044	0.040
Efficiency, %	0.83	0.80	0.71	0.56

**Table 7 polymers-15-02739-t007:** Achieved force by PVA hydrogel P7B2 for each AC voltage and frequency.

Material	PVA Hydrogel P7B2							
Voltage, V	90		110		150		200	
Frequency, Hz	50	500	50	500	50	500	50	500
Activation force, kg	0.460 ± 0.020	0.483 ± 0.094	0.665 ± 0.055	0.704 ± 0.082	1.428 ± 0.042	1.633 ± 0.065	2.032 ± 0.086	2.375 ± 0.045
Required time, s	4.18 ± 0.62	2.55 ± 0.30	3.88 ± 0.52	2.45 ± 0.35	3.58 ± 0.25	1.90 ± 0.18	3.22 ± 0.12	1.50 ± 0.15

**Table 8 polymers-15-02739-t008:** Electrical efficiency of PVA hydrogel P7B2 at conditions of 110 V, 500 Hz, and a load of 50 g.

Material	P7B2
Current, A	0.070 ± 0.003
Power, Watt	7.70 ± 0.10
Activation displacement, %	115 ± 5.6
Activation time, s	3.0 ± 0.2
Power of actuator, Watt	0.19
Efficiency, %	2.44

## Data Availability

The data that support the findings of this study are openly available in the Russian Science Foundation at [https://rscf.ru/project/22-79-10348/ (accessed on 29 May 2023)], reference number 22-79-10348. The authors confirm that the data supporting the findings of this study are available within the article and its Appendix A.

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
