# Peer review of "Preparation of Linear Actuators Based on Polyvinyl Alcohol Hydrogels Activated by AC Voltage"

_polymers, 2023, doi:10.3390/polym15122739_

Round 1

Reviewer 1 Report

1- It is required to explain in summary about "linear actuators" in the introduction.

2- Is there any replacement for the Sodium Tetraborate? what is the solution for the side effects of Sodium Tetraborate such as headache, fever, nausea, dizziness, and weakness?

3- Line 164: the PVA is dissolving in the water at 70 C, why in this study 140 C is used?

4-Line 220: what is a "SUP 1" load cell? Add more details of the experiment to Figure 4.

5- Figure 6 needs replication. try to repeat the experiment three times and show the error bars on the graph.

6- The following reference can be used to improve the introduction :

Sabbagh, F., & Kim, B. S. (2023). Ex Vivo Transdermal Delivery of Nicotinamide Mononucleotide Using Polyvinyl Alcohol Microneedles. Polymers15(9), 2031.

Sabbagh, F., Khatir, N. M., & Kiarostami, K. (2023). Synthesis and Characterization of ?-Carrageenan/PVA Nanocomposite Hydrogels in Combination with MgZnO Nanoparticles to Evaluate the Catechin Release. Polymers15(2), 272.

7- The number of keywords is 8, try to reduce to no more than 6 keywords.

  •  

There are some typo errors in the text. Need for miner edition of the text.

Author Response

1. It is required to explain in summary about "linear actuators" in the introduction.

Thank you for your comment. Please, check the added paragraph:

“In contrast to the circular motion of a traditional electric motor, a linear actuator is an actuator that generates motion in a straight line. Machine tools and industrial machinery, computer peripherals, such as disk drives and printers, valves, and dampers, artificial muscles, and numerous more applications requiring linear motion all use linear actuators. For example, cylinders, whether hydraulic or pneumatic, naturally provide linear motion. In the reference [32], the authors created a system that combines the benefits of rigid and soft robotics by creating an affordable, dependable, muscle-like linear soft actuator used antagonistically to operate a stiff 1-DoF joint. In the reference [33], the authors prepared reverse pneumatic artificial muscle (rPAM), which is made from silicone rubber that is radially constrained by symmetrical double-helix threading. Moreover, spiral artificial muscles based on polyethylene and nylon fibers are also considered one type of linear actuator, which are capable of reversible contraction/extension deformation under heating/cooling cycles [34,35]”.

2. Is there any replacement for the Sodium Tetraborate? what is the solution for the side effects of Sodium Tetraborate such as headache, fever, nausea, dizziness, and weakness?

PVA can be crosslinked using different crosslinking agents, such as glutaraldehyde, formaldehyde, and sulfosuccinic acid.

Generally, borax is considered an available and cheap material with low toxicity, but high exposure to borax can cause headaches, dizziness, lightheadedness, and passing out. However, in our suggested actuators, the PVA/borax hydrogels are filled into a stretchable elastomeric shell as an isolated material, which means that the prepared PVA hydrogels will not be in direct contact with human tissues. In addition, in order to prepare a sufficient actuator with a maximum swelling/shrinking activation, the stretchable elastomeric shell should be tightly closed, which means the evaporated water or any gases will not leak out of the elastomeric shell. Moreover, the amount of borax used was too small.

In addition, there are different crosslinking methods, such as freeze-thaw, cast-drying, theta-gelation, etc. Using borax in our work is also related to its chemical structure, Na2H20B4O17, which gives the possibility to improve the electrical conductivity and mechanical properties of the prepared hydrogels, improve the resistance to stress cracking, and decrease the hydrogels flexibility.

3. Line 164: the PVA is dissolving in the water at 70 C, why in this study 140 C is used?

Please be informed that since the dissolving temperature of PVA depends on the PVA content, its molecular weight, and stirring speed, the heating temperature was increased up to 140 ⁰C in order to decrease the dissolving time (1 h maximum) and minimize the formation of air bubbles as a result of stirring in the prepared hydrogels.

4. Line 220: what is a "SUP 1" load cell? Add more details of the experiment to Figure 4.

Please be informed that "SUP 1" load cell was corrected to “Single Point Parallel Beam Loadcell (SUP 1)” [https://www.made-in-china.com/showroom/amyswj/product-detailCeAJfyaPgHkb/China-Single-Point-Parallel-Beam-Loadcell-SUP1-.html]. Kindly, check the redacted Figure 4.

5. Figure 6 needs replication. try to repeat the experiment three times and show the error bars on the graph.

Please be informed that all experiments were carried out at least three times, and standard deviations were presented in Tables S1–S8 (supplementary materials). Kindly take into consideration that the addition of error bars to the linear curves in Figure 6 will make it too complicated for readers. Please be informed that this information was mentioned in the title of Figure 6 as follows:

Figure 6. Overall extension/contraction of PVA hydrogels reinforced by (a) spiral weave and (b) fabric woven braided mesh under different values of AC-voltage, frequency and a load of 20 kPa. Standard deviations were presented in Tables S1–S8”.

6. The following reference can be used to improve the introduction :

Sabbagh, F., & Kim, B. S. (2023). Ex Vivo Transdermal Delivery of Nicotinamide Mononucleotide Using Polyvinyl Alcohol Microneedles. Polymers, 15(9), 2031.

Sabbagh, F., Khatir, N. M., & Kiarostami, K. (2023). Synthesis and Characterization of ?-Carrageenan/PVA Nanocomposite Hydrogels in Combination with MgZnO Nanoparticles to Evaluate the Catechin Release. Polymers, 15(2), 272.

Please be informed that the suggested references were added to the manuscript.

7. The number of keywords is 8, try to reduce to no more than 6 keywords.

Please be informed that the number of the keywords was reduced as follows:

“polyvinyl alcohol; hydrogel; actuator; AC-voltage; deformation; activation time”.

There are some typo errors in the text. Need for miner edition of the text.

The authors apologize for typos/spelling errors. We have tried to fully rectify them.

Reviewer 2 Report

This work is very interesting and provides one novel way of designing soft actuators. The paper is well-written, and the conclusions are supported by all the data. I am in favor of publication of this manuscript. However, there are still some comments that I would recommend the authors to consider and modify the manuscript accordingly:

1.         When the water evaporates during activation, the resistance of PVA hydrogels is expected to increase, and finally the hydrogel may even become insulating if more and more liquid water changes into gaseous water. Is there a steady state for the actuators shown in this paper? For example, if authors apply 110V, 500Hz voltage to the actuators for 1 minute, could the actuators keep one level of deformation and force for 1 minute as well?

2.        In Figure 4(a), the activation strain is actually the activation displacement. The authors should specify how they calculate the activation strain according to the activation displacement.

3.        Tables 3-6 show the electrical efficiency of different actuators. However, how to measure the actuators efficiency is not mentioned in the paper. In Figure 4, the measurement is only for PVA hydrogels, not for the actuators.

4.        At line 303, the authors said that the frequency has little effect on the deformation. However, in Table S1-S8, there are some data from which we see the frequency has significant effect on the deformation, e.g., Fabric based actuator in Table S3 and S4. Could the authors explain why they have this statement in the main paper?

5.        Since the hydrogel volume change comes from the temperature increase, what is the advantage of this approach compared to using an electric heater to heat up the hydrogel?

6.        For soft actuators, the recovery time after activation is also important, especially for heat-driven actuators. Could the authors provide some data for the recovery time of their actuators?

7.        The font size in Figure 8 is not consistent. The legend of Figure 8(b) looks smaller.

Author Response

This work is very interesting and provides one novel way of designing soft actuators. The paper is well-written, and the conclusions are supported by all the data. I am in favor of publication of this manuscript. However, there are still some comments that I would recommend the authors to consider and modify the manuscript accordingly:

Thank you very much for your kind comments.

  1. When the water evaporates during activation, the resistance of PVA hydrogels is expected to increase, and finally the hydrogel may even become insulating if more and more liquid water changes into gaseous water. Is there a steady state for the actuators shown in this paper? For example, if authors apply 110V, 500Hz voltage to the actuators for 1 minute, could the actuators keep one level of deformation and force for 1 minute as well?

Thank you for your interesting question. By applying an AC-voltage and frequency (110V, 500Hz for example), the actuator can hold the level of deformation and force for up to 20 sec (less than 1 minute); after that, the actuator starts to relax. This is one of the cons that needs to be investigated in detail. Please note that we added this information to the revised manuscript.

  1. In Figure 4(a), the activation strain is actually the activation displacement. The authors should specify how they calculate the activation strain according to the activation displacement.

Thank you for your comment. The authors apologize for this mistake. Please, be informed that the word “strain” was changed to “displacement”.

  1. Tables 3-6 show the electrical efficiency of different actuators. However, how to measure the actuators efficiency is not mentioned in the paper. In Figure 4, the measurement is only for PVA hydrogels, not for the actuators.

Please be informed that in the section "2.4.4. Measurement of activation force of PVA hydrogels", the equations used for calculating the electrical efficiency of both actuators and PVA hydrogels were mentioned. Moreover, kindly check the added information:

“It should be noted that the useful mechanical power generated by the actuator was calculated using the determined values of contraction/extension deformation presented in Tables S1–S8 for 1 m of actuator length”.

  1. At line 303, the authors said that the frequency has little effect on the deformation. However, in Table S1-S8, there are some data from which we see the frequency has significant effect on the deformation, e.g., Fabric based actuator in Table S3 and S4. Could the authors explain why they have this statement in the main paper?

Thank you for your comment. Please be informed that we focused on the activation time more than the deformation values. Moreover, we found that the values of the applied AC-voltage had more influence on the deformation values than the applied frequencies. However, in the conclusion, we have mentioned that by increasing AC-voltage, frequency, and polymer concentration, the overall extension/contraction was increased, and the activation time was decreased, Figure 10.

Please, check the reformulated paragraph:

“It also should be noted that since the increase in the frequency values led to an increase in the deformation values, it is more important that the increase in the frequency values had a significant effect on the activation time of the actuators, which led to a decrease in the activation time by increasing the frequency by ten times”.

  1. Since the hydrogel volume change comes from the temperature increase, what is the advantage of this approach compared to using an electric heater to heat up the hydrogel?

Please be informed that the activation of the stretching/compression processes of actuators generally occurs due to the use of hot and cold mediums (air, water, etc.) for heating and cooling the actuator’s materials, respectively. The disadvantage of this method is that when using a hot/cold medium, the heating/cooling rate on the surface and inside the polymeric material of the actuator can differ significantly. This leads to an uneven distribution of temperature over the polymer volume, which is accompanied by overheating of the surface and the beginning of surface activation before the activation of the inner part of the actuator, which in turn leads to an increase in the time of exposure to a hot/cold medium until the volume sample is completely stretched/contracted. Moreover, using an external hot medium will lead to drying the external surface of the hydrogel. By passing an electric current through the hydrogel, the activation time for this type of heating is noticeably lower than in the case of external bulk heating.

  1. For soft actuators, the recovery time after activation is also important, especially for heat-driven actuators. Could the authors provide some data for the recovery time of their actuators?

Thank you very much for your valuable comment. Kindly take into consideration that the recovery time after activation is almost equal to the activation time (please see videos S1–S5). This is related to the excellent water absorption of the PVA hydrogels and their high adaptability in boiling water as was mentioned in the introduction section. However, the recovery time strongly depends on the applied load, the number of contraction/extension cycles, and the type of reinforcement. It was noticed that after a lot of contraction/extension cycles, the recovery time was significantly increased up to 2 times in comparison with the activation time, which can be related to the high heat capacity of water, which in turn needs more investigation to maintain the level of the recovery time. Please be informed that this information has been added to the manuscript.

  1. The font size in Figure 8 is not consistent. The legend of Figure 8(b) looks smaller.

Please, check the redacted Figure 8.
